# Comparing Remote Speckle Plethysmography and Finger-Clip Photoplethysmography with Non-Invasive Finger Arterial Pressure Pulse Waves, Regarding Morphology and Arrival Time

**DOI:** 10.3390/bioengineering10010101

**Published:** 2023-01-11

**Authors:** Jorge Herranz Olazabal, Fokko Wieringa, Evelien Hermeling, Chris Van Hoof

**Affiliations:** 1IMEC, 3000 Leuven, Belgium; 2Faculty of Engineering Science, Katholieke Universiteit Leuven (KUL), 3000 Leuven, Belgium; 3IMEC NL, 5656 AE Eindhoven, The Netherlands; 4Division of Internal Medicine, Department of Nephrology, University Medical Center Utrecht, 3584 CX Utrecht, The Netherlands

**Keywords:** camera-based, speckle contrast analysis, optical monitoring, laser speckle, PPG, SPG

## Abstract

Objective: The goal was to compare Speckle plethysmography (SPG) and Photoplethysmography (PPG) with non-invasive finger Arterial Pressure (fiAP) regarding Pulse Wave Morphology (PWM) and Pulse Arrival Time (PAT). Methods: Healthy volunteers (n = 8) were connected to a Non-Invasive Blood Pressure (NIBP) monitor providing fiAP pulse wave and PPG from a clinical transmission-mode SpO_2_ finger clip. Biopac recorded 3-lead ECG. A camera placed at a 25 cm distance recorded a video stream (100 fps) of a finger illuminated by a laser diode at 639 nm. A chest belt (Polar) monitored respiration. All signals were recorded simultaneously during episodes of spontaneous breathing and paced breathing. Analysis: Post-processing was performed in Matlab to obtain SPG and analyze the SPG, PPG and fiAP mean absolute deviations (MADs) on PWM, plus PAT modulation. Results: Across 2599 beats, the average fiAP MAD with PPG was 0.17 (0–1) and with SPG 0.09 (0–1). PAT derived from ECG–fiAP correlated as follows: 0.65 for ECG–SPG and 0.67 for ECG–PPG. Conclusion: Compared to the clinical NIBP monitor fiAP reference, PWM from an experimental camera-derived non-contact reflective-mode SPG setup resembled fiAP significantly better than PPG from a simultaneously recorded clinical transmission-mode finger clip. For PAT values, no significant difference was found between ECG–SPG and ECG–PPG compared to ECG–fiAP.

## 1. Introduction

This work focuses on comparing both Speckle plethysmography (SPG) and Photoplethysmography (PPG) with non-invasive Finger Arterial Pressure (fiAP) regarding Pulse Wave Morphology (PWM) and respiration-induced Pulse Arrival Time (PAT) modulation.

SPG is a technology that is derived from Laser Speckle Contrast Imaging (LSCI) [1] by analyzing variations in the speckle contrast. The formation of laser speckles depends on photonic interference. A laser beam illuminating a static object will produce a nearly static speckle pattern (only varying with Brownian motion and laser coherence). However, for living tissue, any movement of scattering or reflecting tissue components causes a modulation in the speckle pattern. Previous literature on the subject claimed SPG as a purely flowmetric technology [2], but the supporting evidence was solely collected from experiments where flow was variated linearly in a rigid plastic tube.

Although this formed a solid investigation to show that flow modulations correlate well with SPG [2,3,4], more investigation is needed to claim that the SPG waveform *solely* depends on blood flow: human blood vessels are flexible, and pressure waves cause the blood vessel walls to expand and contract, thereby compressing the surrounding tissue. This movement will also be perceived by SPG, additionally to the streaming blood cells and should not be neglected. Hence, we studied the influence of blood pressure pulse waves.

The most accurate way of measuring dynamic blood pressure is invasive (via a catheter) [5]. Blood pressure can also be estimated using non-invasive methods [6]. The most used clinical method is a sphygmomanometer, in which an inflatable cuff is positioned around a part of the body, typically the upper arm, and blood pressure is estimated by auscultation or the oscillometric approach [7]. Although sphygmomanometers are widely used in daily clinical practice, they have the disadvantage of providing discontinuous snapshot measurements that can substantially deviate from actual blood pressure [8]. Alternatively, blood pressure can be measured by using the volume clamp method of Peñáz [9]. This method uses a feedback loop that dynamically pressurizes a finger cuff to clamp optical transmission. The dynamic cuff pressure that keeps the optical finger transmission constant provides the finger Arterial Pressure (fiAP) pulse wave. This device provides a continuous blood pressure waveform, but it is bulky and can only be applied for a limited amount of time. 

Upcoming non-invasive methods use bio-signals, usually PPG, and assess blood pressure through Pulse Wave analysis PWA [10], Pulse Arrival Time (PAT) analysis [11], Pulse Wave Velocity (PWV) analysis [12] or Pulse Transit Time analysis [13]. Although these PAT- and PWA-based methods have the potential advantage of allowing continuous monitoring of blood pressure, they can be substantially biased as PWA and PAT might change independently of actual blood pressure [14], e.g., due to differences in vasomotor tone.

This study aims to compare simultaneously recorded SPG and PPG waveform morphologies with the pressure waveform measured by a clinical Finger Arterial Pressure (fiAP) monitor. In addition, this study also compares ECG–fiAP PAT against ECG–PPG and ECG–SPG PAT measurements. With this, we intend to shed light on the possible use of SPG for blood pressure estimation.

## 2. Materials and Methods

The reference device used as Non-Invasive Blood Pressure (NIBP) waveform monitor was Finapres Nova [15]. The device also incorporates a clinical transmission-mode SpO_2_ finger clip with output of the near-infrared PPG pulse wave. 

A Biopac MP160 was used to record reference ECG, respiration from a chest belt (Polar), the PPG and fiAP signals coming from Finapres NOVA and the common trigger pulses for the laser and camera (for synchronization).

A camera (acA2000-340 km, Basler, Breda, Netherlands) was placed at 25 cm distance to record 100 Laser Speckle Contrast Images per second of a finger being illuminated by a laser diode at a wavelength of 639 nm (HL6358MG, Thorlabs, Dortmund, Germany). The approximated coherence length of the laser diode was 0.9 mm [16]. Camera video recordings were stored on a separate computer and post-processed to produce the SPG signal. Synchronization among camera, laser diode and Biopac recordings was achieved by a microcontroller (STM32, STMicroelectronics, Diegem, Belgium) producing the common central sync pulses.

The study protocol was approved by the institutional ethical committee and device safety board. We recruited 8 healthy volunteers and connected them to the setup as shown in Figure 1. The study population was composed of healthy subjects between the ages of 23 and 55 years. The gender distribution of the participants was 6 males and 2 females. Measured breathing-induced changes in systolic values of BP as measured by NIBP fiAP ranged from Systolic High 145(±36) mmHg to Systolic Low 106(±34) mmHg.

All signals (including synchronization pulses from the microcontroller) were recorded simultaneously (Biopac, Biopac Systems, Inc., Varna, Bulgaria) during episodes of spontaneous breathing (3 min) as well as paced breathing (3 min, at 6 breaths/min) and analyzed for morphology mean absolute deviation (MAD) and PAT modulation (using ECG–fiAP PAT as reference, with ECG R-peaks as gold standard for beat-start timing). During the experiments, the operator and volunteers wore laser safety glasses. Figure 1 shows the schematic of the experimental setup, depicting the connections between the experimental and reference devices, and showing the connection of the participants to the system. No specific movement artifact suppression algorithms were applied, which is why during the recordings, subjects were asked to keep their hand as still as possible, resting on a 3D-printed ergonomic support.

## 3. Analysis

The video sequences were processed to obtain SPG using a standard deviation sliding window (sized 7 × 7 pixels) to scan each video frame, following the equation [16]: SPG (frame) = mean [σ_7×7_ (frame)](1)

The objective of the data analysis was to compare waveform morphology and a temporal feature (PAT) of finger-clip PPG and reflective SPG against a continuous finger Arterial Pressure reference waveform (fiAP). 

### 3.1. Morphology Analysis

The morphological analysis consisted of the following steps (see also Figure 2): First, upstrokes were calculated from PPG, SPG and reference signal (fiAP). Then, PPG, SPG and reference signal (fiAP) were normalized beat-to-beat, by the maximum and minimum of each beat. After this, PPG as well as SPG were subtracted from the normalized fiAP signal to obtain respective PPG and SPG absolute deviations. Finally, the signal difference of PPG and SPG with fiAP was averaged for each beat, producing MAD values. For visualization purposes, boxplots of MAD beat-averaged were generated for PPG as well as for SPG (Section 4). 

MAD values provide an objective absolute calculation of the morphological difference among different signals. It is a convenient method to compare signals coming from different devices with similar morphology.

### 3.2. Pulse Arrival Time Analysis

The temporal feature chosen for the secondary analysis was Pulse Arrival Time (PAT). This variable was calculated by measuring the time elapsed between the ECG R-peak and the foot of the fiAP (used as reference), PPG and SPG signals. Figure 3 depicts the calculation of PAT from ECG–PPG vs PAT from ECG–SPG, and the same calculation was also performed for ECG–fiAP. This calculation was performed by measuring the time elapsed between the R peak of ECG and the valley of their respective PPG, SPG and fiAP signals.

Offset values of PAT are not meaningful because the signals are obtained from different devices, which implies that they have different time delays. However, the PAT *variations* should be comparable (e.g., related to respiration).

### 3.3. Statistical Analysis

For the morphological analysis, MAD values were calculated beat-to-beat from PPG as well as SPG signals for all subjects. We performed a paired *t*-test for testing the null hypothesis that MAD PPG and MAD SPG are from populations with equal means at the 5% significance level. This would demonstrate whether the difference between MAD values inter-subject is significant for PPG versus SPG. Additionally, we performed two more *t*-tests to check if the difference in MAD mean values was significantly different between periods of *spontaneous* versus *paced* breathing for PPG and SPG.

For the temporal analysis, PAT values were calculated beat-to-beat from PPG, SPG and fiAP for all subjects. To analyze the similarity between fiAP and the signals under investigation, intra-subject correlation coefficient was calculated between fiAP and PPG and compared with the correlation coefficient between fiAP and SPG. Finally, an average correlation coefficient was calculated for all subjects.

## 4. Results

Figure 4 shows an overview of the measured signals from one subject during spontaneous (up to ~105 s) and paced breathing (after ~105 s). In this figure, variations in all signal traces are clearly appreciated between periods of spontaneous and paced breathing. It is important to notice that these variations differ for each signal. Note that paced breathing-induced changes in blood pressure are clearly depicted in the PPG–DC values and SPG–DC values. In addition, paced breathing leads to variations in PAT, observed for all signals (fiAP, SPG and PPG). The analysis of the PAT results is contained in Section 4.2.

### 4.1. Morphology Results: Pulse Waveform

In total, 2599 beats were analyzed (total of 8 volunteers). A visual inspection of the SPG, PPG and fiAP morphologies revealed a higher resemblance of the SPG waveform with the fiAP waveform compared to that of PPG, see Figure 5. An analysis confirmed that, compared to fiAP, the MAD of PPG 0.17 (range 0–1) was bigger than the MAD of SPG 0.09 (range 0–1).

We performed a paired *t*-test (*t*-test2 on Matlab, version R2022a) to test the null hypothesis that the MAD values from PPG and SPG are from populations with equal means. The *t*-test showed that the difference in the mean between SPG and PPG MAD is significant (*p* < 0.0001), rejecting the null hypothesis at the alpha significance level. 

To check whether the similarity in morphology was not dominantly present for spontaneous or paced breathing, we repeated the morphology analysis discriminating between episodes of spontaneous and paced breathing, see Figure 6.

Separate paired *t*-tests on MAD PPG and MAD SPG between episodes of spontaneous and paced breathing did not reveal a significant difference, neither for SPG nor for PPG. Both for spontaneous and paced breathing, the remote SPG morphology remained significantly more similar to fiAP than contact PPG.

### 4.2. Temporal Results: Pulse Arrival Time (PAT)

Table 1 shows remote SPG PAT and contact PPG PAT correlation coefficients with fiAP PAT averaged per subject, from all eight volunteers. The paired *t*-test results show no significant difference between the PAT correlation coefficients derived from contact PPG and remote SPG when comparing to the fiAP reference: PAT fiAP vs PAT PPG (mean 0.673 ± 0.46) and PAT fiAP vs PAT SPG (mean 0.658 ± 0.47). 

Average peak-to-foot amplitudes were calculated for SPG and PPG as a surrogate of signal quality between subjects, shown as AC (Alternating Component) PPG (mean 1.25) and AC SPG (mean 0.0084) in Table 1. Subjects 3 and 6 showed significantly lower average peak-to-foot values, which is an indication of a low signal quality on these subjects. This has a direct impact in the correlation coefficient listed for these two subjects in Table 1.

Even if we discarded the two subjects with the poorest peak-to-foot values (marked red in Table 1), the results would not show a significant difference for PAT fiAP vs. PAT contact PPG (0.75) and for PAT fiAP vs. PAT remote SPG (0.79).

## 5. Discussion

This study demonstrates that camera-derived remote SPG has a better resemblance with the fiAP waveform than contact finger-clip PPG. In addition, PAT of remote reflective SPG shows results comparable with contact transmissive PPG. This is an important finding because PWA and PAT are widely being investigated for non-invasive BP assessment [10,11,17], especially within wearable technologies [18]. 

The resemblance of the PPG waveform to the BP waveform constitutes the foundation for PWA methods [10,17]. Given the observed *bigger* resemblance of SPG morphology to continuous blood pressure, it is logical to expect that remote SPG may make a difference for non-invasive BP assessment. In addition, SPG might benefit from measurements at a shorter distance to improve accuracy, which could mean that its implementation into wearable technologies could bring advantages for non-invasive BP assessment and other hemodynamic metrics. 

PPG PAT has been investigated as a marker for arterial stiffness and BP assessment [11,19]. *Remote reflective* SPG has shown results comparable to *contact transmissive* PPG, which might allow for PAT calculations at body locations where the PPG quality is compromised, at body locations where contact should be avoided or simply to improve comfort. This investigation was performed using a finger-clip transmissive PPG sensor. In wearable technologies, reflective measurements are preferred (as they allow greater freedom at the measurement site) but have a lower SNR for PPG, whereas SPG *excels* in reflection mode [16]. 

Within the literature, the PPG waveform is often regarded as a proxy of the local arterial blood pressure waveform, and the origin of PPG is mostly explained by the pulsatile volumetric dilatation of the arterial system [20]. In addition, the literature mostly regards the SPG waveform as a proxy of local blood flow, and the origin of SPG is mostly explained as the speckle modulation caused by pulsatile blood flow [2]. If these two assumptions were fully correct, one would expect the PPG pulse wave to resemble the blood pressure pulse wave morphology more closely than SPG. Yet, we observed the opposite behavior.

In previous work on SPG [2], the utilized system applied a contact probe with the transmission mode, with presumably a high (but not disclosed) coherence length. In contrast, the system utilized in this work applied a remote camera in the reflective mode, combined with a short coherence length laser diode. Due to the short coherence length, the system described in this work might be more sensitive to movements of the tissue surface (which would correlate with tissue movement from arterial pulsation [21]) and capillary flow. Capillaries are vessels with low terminal resistance and low compliance (barely any dilation [22]), which means that they show a linear relationship between pressure and flow. On the other hand, arterioles have high terminal resistance and are equipped with sphincters, which makes them able to shunt or dilate to regulate blood flow. These vessels have high compliance, which gives an out-of-phase relationship between flow and pressure [23].

Both types of vessels could have different impacts on SPG, and this would give a different explanation to SPG’s origin, depending on the location (capillaries are more superficial in skin than arterioles and arteries [24]).

Another interesting fact is that the applied NIBP fiAP reference device uses an optical probe in the transmission mode (highly resembling PPG) to measure continuous BP in the finger [25]. Nevertheless, when comparing the fiAP signal with a transmissive PPG signal and SPG obtained in the reflective mode, fiAP morphology was proven to have a significantly closer resemblance to SPG.

In our present study, we discovered no significant difference between contact PPG and remote SPG for PAT measurement. Nevertheless, in a previous study, we found that *remote* SPG can be at least as accurate as *contact* PPG for beat-to-beat interval measurement when comparing it to ECG [26]. PAT from reflective SPG at an improved Technology Readiness Level, using a miniaturized measurement setup, might deserve more investigation. Clearly, embodiments with cheap mass-produced laser sources and miniature cameras would be more attractive from an economic point of view.

Movement artifacts and their suppression mechanisms were not investigated during this study. Yet, it is known that both PPG and SPG are sensitive to movement artifacts [27], especially in wearable monitoring. Recent developments in flexible electronics to reduce the relative movement between probe and skin might offer advantages [27,28]. Rein et al. [28] demonstrated stretchable conformable lightweight optical incoherent emitter–detector pairs, providing improved wearability and movement artifact reduction for PPG. For SPG, however, the application of miniature camera modules (such as those used in mobile phones) might be a simpler approach.

## 6. Conclusions

In this work, we compared a standard contact (finger clip) *transmissive* PPG device and an experimental camera-derived non-contact *reflective* SPG setup, to the fiAP waveform from a clinical NIBP monitoring reference device. Both pulse wave morphologies and PAT modulations were compared. 

In contrast to the expected behavior, our findings reveal that the contactless reflection mode SPG (from an *experimental* setup, using a laser with a coherence length of 0.9 mm) showed a significantly higher resemblance to fiAP regarding waveform morphology than PPG from a *clinical* transmission-mode finger clip. A correlation between PAT derived from SPG was found similar to PAT from PPG when compared to fiAP with the present setup. 

The realization of an improved probe that would combine SPG and PPG in the same device, deriving both signals from the *same photons,* seems a logical next step to investigate how to harvest the best of both modalities in order to advance the field of peripheral circulation monitoring.

## Figures and Tables

**Figure 1 bioengineering-10-00101-f001:**
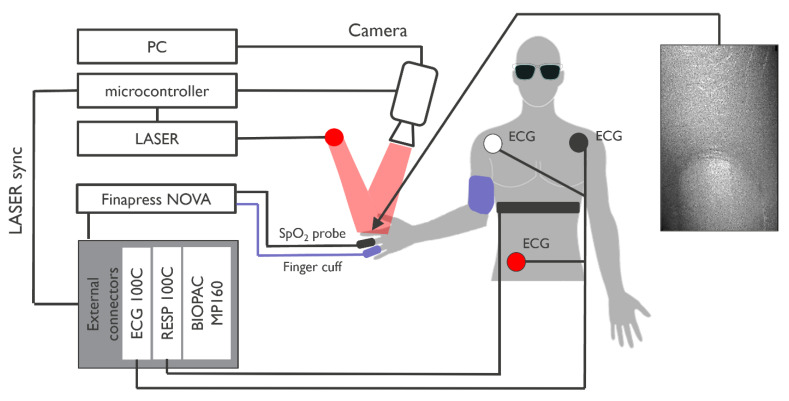
Block diagram of the experimental setup with example of image captured by the system (320 × 512 px).

**Figure 2 bioengineering-10-00101-f002:**
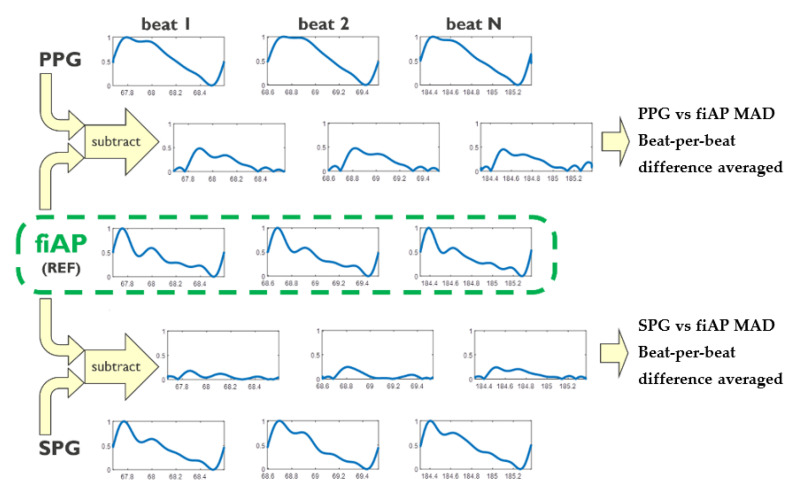
Comparison of PPG (**top**) and SPG (**bottom**) waveform morphologies with non-invasive Finger Arterial Pressure (fiAP) reference. For boxplots of the resulting PPG vs. fiAP and SPG vs. fiAP see Section 4.

**Figure 3 bioengineering-10-00101-f003:**
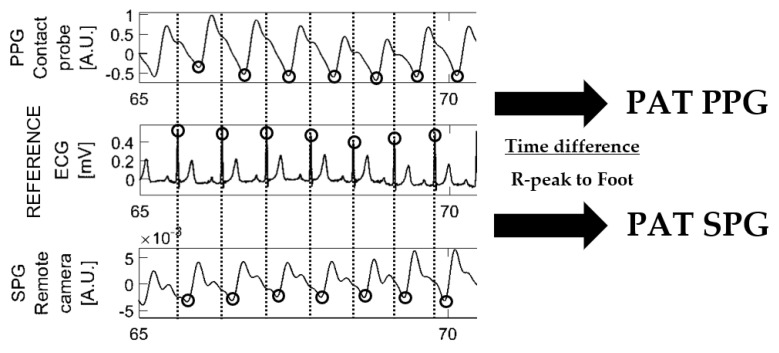
Example of Pulse Arrival Time (PAT) calculation on PPG and SPG.

**Figure 4 bioengineering-10-00101-f004:**
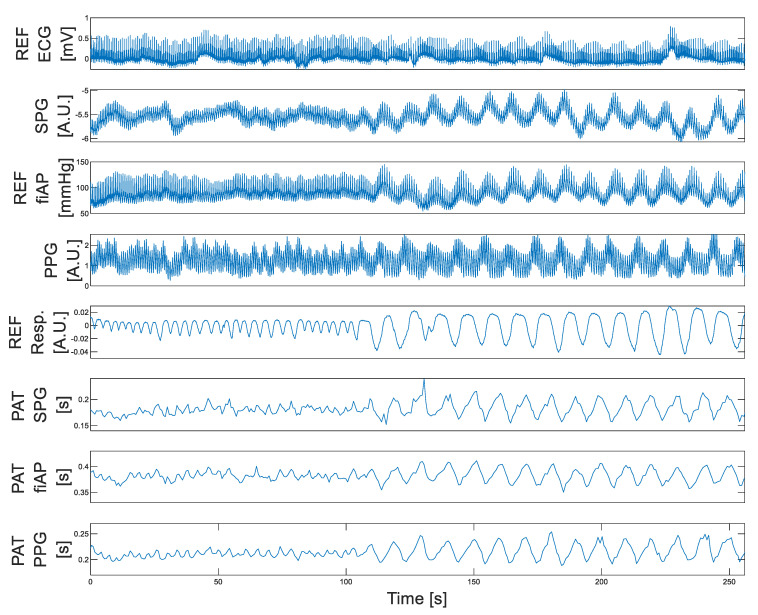
Spontaneous versus paced breathing. Example of signals collected from subject 1. Paced breathing starts at ~105 s. Parameters from top to bottom: ECG, SPG, fiAP, PPG, Respiration, PAT SPG, PAT fiAP and PAT PPG. Note: Time base was chosen to highlight amplitude modulation and correlation of PAT (across > 250 heartbeats). Individual heartbeats appear as spikes, modulated in the respiration rhythm. For SPG, PPG and ECG at a faster time base, see Figure 3.

**Figure 5 bioengineering-10-00101-f005:**
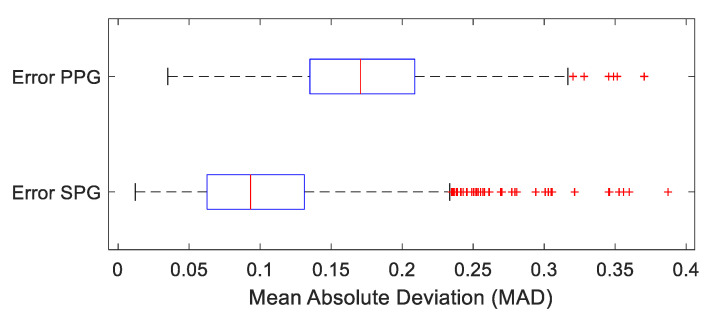
Beat-averaged MAD of fiAP vs. PPG (top, average MAD 0.17) and fiAP vs. SPG (bottom, average MAD 0.09).

**Figure 6 bioengineering-10-00101-f006:**
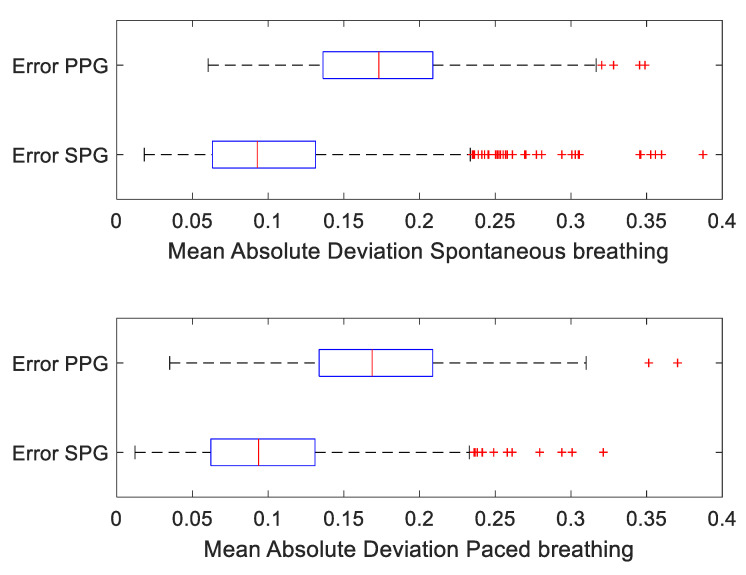
MAD beat-averaged of fiAP vs. PPG and fiAP vs. SPG, differentiating between episodes of spontaneous breathing and paced breathing.

**Table 1 bioengineering-10-00101-t001:** R (correlation) of fiAP and PPG PAT vs. R of fiAP and SPG PAT. Red highlight shows signals below the 0.006 quality threshold.

Subject	R PPG PAT	AC PPG	R SPG PAT	AC SPG
1	0.86	1.16	0.89	0.0089
2	0.58	1.33	0.71	0.0088
3	0.21	1.51	0.18	0.0049
4	0.84	1.08	0.63	0.0078
5	0.83	1.06	0.83	0.0086
6	0.65	0.99	0.35	0.0041
7	0.51	1.22	0.73	0.0113
8	0.9	1.66	0.95	0.0129
**Average**	**0.6725**	**1.25**	**0.65875**	**0.0084**

## Data Availability

Not applicable.

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
