# Peer review of "Comparing Remote Speckle Plethysmography and Finger-Clip Photoplethysmography with Non-Invasive Finger Arterial Pressure Pulse Waves, Regarding Morphology and Arrival Time"

_bioengineering, 2023, doi:10.3390/bioengineering10010101_

Round 1
Reviewer 1 Report
The manuscript by Olazabal et al. demonstrates the comparison of SPG and PPG methods with finger arterial pressure in the context of PAT and PWM. The work is interesting and technically sound. It is well written and will catch the interest of readers. Below are some points to be considered in the revision.
1. How do authors deal with noise associated with human actions/movements/activities? Which signal processing strategies were implemented or which assumptions were made? Particularly in the SPG method it seems there is a huge source of noise. It should be clearly mentioned in the manuscript.
2. It is important including state of the art studies in wearable electronic research as a future path of SPG or PPG integrated methods. Particularly, recent developments in flexible electronics technology would be suitable to mention. For example, below approaches provide unique SPG and PPG tools which can be discussed in the manuscript as a future direction.
a) Rein, M., et al. Diode fibres for fabric-based optical communications. Nature 560, 214–218 (2018).
b) Webb, R.C., et al. Epidermal devices for noninvasive, precise, and continuous mapping of macrovascular and microvascular blood flow. Science advances, 1, p.e1500701 (2015).
3. English check required. Title can be rewritten; with current version it is lengthy and difficult to understand.
Author Response
- How do authors deal with noise associated with human actions/movements/activities? Which signal processing strategies were implemented or which assumptions were made? Particularly in the SPG method it seems there is a huge source of noise. It should be clearly mentioned in the manuscript.
Subjects were asked to keep the hand as still as possible during the recordings, to avoid movement artifacts. The only signal processing step before the morphological analysis is the obtention of the SPG signal by using a sliding standard deviation window (7 by 7 pixels). SPG showed little amount of noise, to clarify figure 4 shows a time window of more than 250s, when depicting signals in this manner individual heartbeats might seem like noise. Figure 3 shows typical signals in which you can appreciate very good signal-to-noise ratios on both the SPG and PPG signals.
ADDITION TO THE ARTICLE (methods and materials): No specific movement artifact suppression algorithms were applied, which is why subjects were asked to keep their hand as still as possible during the recordings, resting on a 3D-printed ergonomic support.
We also changed the caption text of figure 4 to pronounce the slow time base which, at a glance, might be perceived as noisy heartbeats instead of respiration induced patterns.
CHANGE CAPTION OF Figure 4: Spontaneous versus paced breathing. Example of signals collected from subject 1. Paced breathing starts at ~105s. Parameters from top to bottom: ECG, SPG, fiAP, PPG, Respiration, PAT SPG, PAT fiAP & PAT PPG. Note: Time base was chosen to highlight amplitude modulation & correlation of PAT (across > 250 heartbeats). Individual heartbeats appear as spikes, modulated in the respiration rhythm. For SPG, PPG and ECG at a faster time base, see Figure 3.
- It is important including state of the art studies in wearable electronic research as a future path of SPG or PPG integrated methods. Particularly, recent developments in flexible electronics technology would be suitable to mention. For example, below approaches provide unique SPG and PPG tools which can be discussed in the manuscript as a future direction.
a) Rein, M., et al. Diode fibres for fabric-based optical communications. Nature 560, 214–218 (2018).
b) Webb, R.C., et al. Epidermal devices for non-invasive, precise, and continuous mapping of macrovascular and microvascular blood flow. Science advances, 1, p.e1500701 (2015).
ADDITION TO THE ARTICLE (Discussion): Movement artifacts and their suppression mechanisms were not investigated during this study. Yet it is known that both PPG and SPG are sensitive to movement artifacts [b], especially in wearable monitoring. Recent developments in flexible electronics to reduce the relative movement between probe and skin might offer advantages [a, b]. Rein et al. [a] demonstrated stretchable conformable light-weight optical incoherent emitter-detector pairs, providing improved wearability and movement artifact reduction for PPG. For SPG, however, the application of miniature camera modules (like used in mobile phones) might be a simpler approach.
- English check required. Title can be rewritten; with current version it is lengthy and difficult to understand.
ORIGINAL TITLE (24 words): A comparison of contactless camera-based Speckle Plethysmography and finger clip Photoplethysmography with Non-Invasive Finger Arterial Pressure, regarding Pulse Wave Morphology and Pulse Arrival Time.
ALTERNATIVE 1 (20 words – 171 characters): Comparing remote Speckle Plethysmography and finger clip Photoplethysmography with Non-Invasive Finger Arterial Pressure Pulse Waves, regarding Morphology and Arrival Time.
ALTERNATIVE 2 (19 words – 134 characters): Comparing remote SPG and finger clip PPG with Non-Invasive Finger Arterial Pressure Pulse Waves, regarding Morphology and Arrival Time.
In the revised manuscript, we now have included Alternative title 1. However, if you would like to shorten it further, we might consider replacing Speckle Plethysmography and Photoplethysmography by their respective acronyms, namely SPG and PPG. In word count this only saves one word, but it significantly reduces the character count. We would be happy with either alternative 1 or 2, and leave this choice up to reviewer 1 and the editor.
Reviewer 2 Report
In current manuscript, the authors compared an experimentally setup contactless SPG to the standard PPG. While technical setup and analysis were valid which lead to the conclusion that SPG can achieve similar performance to a commercial PPG, the significance of this study needs to be greatly elaborated. From the application point of view, a functional SPG with expensive camera and optics setup doesn't seem like a logical nor economical alternative to PPG given both techniques were non-invasive. In addition, the marginally better fiAP correlation from SPG using an experimentally fine-tuned system compared to a low cost, readily available PPG device is quite deceiving from cost-performance point of view.
Author Response
The point raised by the reviewer is completely valid. However, cost was not taken into account in this first exploratory study. Our intention was to investigate the best SPG signals we could obtain with an assembly of off-the-shelf components. The readily available PPG and finger arterial pressure systems are clinically accepted devices with cheaper components, but a much higher technology readiness level (that also involved a lot of engineering and optimization steps). In the clinical PPG system, cost-effectiveness has already been highly engineered, but not yet in our system. In the future, cost-effectiveness of SPG system clearly should be highly improved. The target of the article was to investigate the morphological and temporal correlation of fiAP with SPG & PPG signals. Your comments are indeed revealing a need for improvement in the discussion section.
ADDITION TO THE ARTICLE (Discussion line 294): PAT from reflective SPG at an improved Technology Readiness Level, using a miniaturized measurement setup might deserve more investigation. Clearly, embodiments with cheap mass-produced laser sources and miniature cameras would be more attractive from an economic point of view.
Reviewer 3 Report
i am grateful to be afforded the opportunity to review your manuscript.
i found the study sufficiently grounded within existing practice and supported by literature.
the manuscript itself is cogently argued and clearly structured.
i am happy to recommend it for publication in its present form.
Author Response
Thank you for your kind and encouraging words!
Reviewer 4 Report
In the paper Authors compare a stardard transmittive PPG device with custom built reflective devicethat is utilizing laser source and the camera. Authors found that their setup gives better results than traditional PPG with reference to finger arterial pressure.
I think paper is weel written and shows nice results but I would like to clarify few things.
Authors cite 2599 beats few times in the paper. It is a total amount of beats across all the volunteers, amount of beats for single volunteer? Please specify.
The laser-camera system is fragile to person movements or even muscle contractions. Please comment on how the movements of the body vs camera were notified / collected / removed.
Also it would be valuable to share the image of the camera that captures the data with some explanatory images.
General remark - making the wearable device that uses the laser and the camera would be a challenging task.
Remaining part of the paper seem to be clear. Thank You
Author Response
Authors cite 2599 beats few times in the paper. It is a total amount of beats across all the volunteers, amount of beats for single volunteer? Please specify.
ADDITION TO THE ARTICLE (Morphology results line 200): In total 2599 beats were analyzed (total of 8 volunteers).
The laser-camera system is fragile to person movements or even muscle contractions. Please comment on how the movements of the body vs camera were notified / collected / removed.
ADDITION TO THE ARTICLE (methods and materials): No specific movement artifact suppression algorithms were applied, which is why subjects were asked to keep their hand as still as possible during the recordings, resting on a 3D-printed ergonomic support.
ADDITION TO THE ARTICLE (Discussion): Movement artifacts and their suppression mechanisms were not investigated during this study. Yet it is known that both PPG and SPG are sensitive to movement artifacts [b], especially in wearable monitoring. Recent developments in flexible electronics to reduce the relative movement between probe and skin might offer advantages [a, b]. Rein et al. [a] demonstrated stretchable conformable light-weight optical incoherent emitter-detector pairs, providing improved wearability and movement artifact reduction for PPG. For SPG, however, the application of miniature camera modules (like used in mobile phones) might be a simpler approach.
Also it would be valuable to share the image of the camera that captures the data with some explanatory images.
Figure 1 has been modified and now includes an example image.
CAPTION MODIFICATION: Figure 1. Block diagram of the experimental setup with example of image captured by the system (320x512px).
General remark - making the wearable device that uses the laser and the camera would be a challenging task.
We agree that this will be a challenging task, but it may be very rewarding, as we demonstrated that in principle there is additional information in the SPG waveform. Therefore, we end the article with the following paragraph: “Realization of an improved probe that would combine SPG and PPG in the same device, deriving both signals from the same photons seems a logical next step to investigate how to harvest the best of both modalities in order to advance the field of peripheral circulation monitoring.”
ADDITION TO THE ARTICLE (Discussion): Movement artifacts and their suppression mechanisms were not investigated during this study. Yet it is known that both PPG and SPG are sensitive to movement artifacts [b], especially in wearable monitoring. Recent developments in flexible electronics to reduce the relative movement between probe and skin might offer advantages [a, b]. Rein et al. [a] demonstrated stretchable conformable light-weight optical incoherent emitter-detector pairs, providing improved wearability and movement artifact reduction for PPG. For SPG, however, the application of miniature camera modules (like used in mobile phones) might be a simpler approach.
